# The Association between COVID-19 Pandemic and the Quality of Life of Medical Students in Silesian Voivodeship, Poland

**DOI:** 10.3390/ijerph191911888

**Published:** 2022-09-20

**Authors:** Szymon Szemik, Maksymilian Gajda, Aleksandra Gładyś, Małgorzata Kowalska

**Affiliations:** 1Department of Nursing Propaedeutics, School of Health Sciences, Medical University of Silesia, 40-027 Katowice, Poland; 2Department of Epidemiology, School of Medicine in Katowice, Medical University of Silesia, 40-055 Katowice, Poland

**Keywords:** quality of life, health, pandemic, COVID-19, medical students, lifestyle, online learning, e-learning

## Abstract

(1) Background: Since the COVID-19 pandemic spread rapidly in 2020, medical universities have been affected by a particular crisis. Due to the increased risk of SARS CoV-2 transmission, the authorities of medical faculties all over the world, including Poland, started to minimize direct contact between students. The objective of the paper is to identify and assess determinants of the quality of life among medical students in Poland before and during the COVID-19 pandemic. (2) Methods: We analyzed data obtained in a cross-sectional study performed among three groups of students tested in three consecutive research periods: period before the COVID-19 pandemic, the COVID-19 pandemic lockdown period and the COVID-19 pandemic period following lockdown. (3) Results: The total number of participants was 1098. We identified that the research period before the COVID-19 pandemic was the risk factor for lower quality of life in terms of the somatic and environmental domains. It was also confirmed that determinants such as poor financial situation, low frequency of physical activity and bad self-declared health status harmed the QoL scores in all domains. (4) Conclusions: The obtained results confirmed that better financial situation, higher physical activity and better self-declared health status were statistically significant factors improving the quality of life of first-year medical students in Poland. The findings of our study also showed that the declared somatic and environmental domains of QoL among medical students were better during the COVID-19 pandemic period. Our observations confirmed that the immediate implementation of e-learning could protect against the deterioration of mental health and quality of life in first-year medical students during possible future epidemic crises.

## 1. Introduction

Since the COVID-19 pandemic spread rapidly in 2020, numerous aspects of life and the organization of societies have changed. The healthcare system has significantly deteriorated, and at the same time, medical universities have been affected by a particular crisis. Due to the increased risk of SARS CoV-2 transmission, the authorities of medical faculties all over the world, including Poland, started to minimize direct contact between students in March 2020 [1]. The expansion of the online learning environment was accelerated, while many academic teachers were delegated to work with COVID-19 patients and their physical presence in laboratories was significantly limited and complicated the education process [2,3].

In general, medical education is considered to be highly demanding. To evaluate the intensity of medical training, it is important to identify factors that could have an impact on students’ learning, mental health, and well-being [4]. The assessment of quality of life (QoL) plays a significant role in the measurement of mental health problem prevalence among physicians. For example, an important issue in this profession is burnout, which has a major negative impact on the quality of life of medical staff, which consequently corresponds to a decrease in the quality of care and leads to an economic burden on the healthcare system [4].

There is evidence that medical students are exposed to a higher level of mental illness and psychological distress, as well as poorer mental well-being in comparison to the general population [5]. Moreover, a review of available published data representing the situation before the COVID-19 pandemic has shown, for instance, the significant relationship between the prevalence of mental health problems among medical students and their personality characteristics [6]. Some published data indicated that primarily neuroticism was associated with a higher level of perceived stress, and symptoms of anxiety and depression [7,8] as well as suicidal thoughts and behaviors [9]. Furthermore, differences in the subjective well-being of medical students may be related to the factors identifying the study environment, such as self-esteem and social support from colleagues at a medical school, partners or other family members. The most important determinants that negatively affect well-being were also the perception of the medical school as an unfriendly environment, concerns about future work and own competencies, as well as perceived exam stress [10] and the pressure to achieve high scores.

Individual behaviors of medical students including so-called lifestyle (diet, physical activity, alcohol drinking, tobacco smoking and the use of psychoactive substances) could be listed among well-known determinants of quality of life [11,12]. For example, our previous paper indicates that about 30.9% of medical students at the Medical University of Silesia were hazardous drinkers [13].

On the other hand, analysis of the impact of the COVID-19 pandemic on the QoL and health of medical students is an important and valid issue. There are many studies available regarding the quality of life and mental health of medical and nursing students that were conducted mostly during a pandemic [14,15,16,17]. Hence, the findings from systematic review and meta-analysis confirmed the prevalence of various mental health problems among nursing students, including sleep disturbances, anxiety, depression, stress and fear [18]. Additionally, it has also been indicated that during the pandemic period nursing students reported a higher level of student-life stress compared to other academic majors [19].

However, there are no studies assessing the influence of the COVID-19 pandemic on the quality of life of medical students in Poland. These circumstances justified the study of the phenomenon of quality of life among students at one of the largest medical universities in Poland, located in Katowice (the total number of students is close to 10,000). We believe that the results of well-planned comparisons between the groups of medical students from the period before the COVID-19 pandemic and after the lockdown will bring insights into how COVID-19 pandemic changes affected the level of quality of life in medical students.

### Objective

The objective of the paper is to identify and assess determinants of the quality of life among medical students of the Medical University of Silesia in Katowice, Poland (MUoS) before and during the COVID-19 pandemic.

## 2. Materials and Methods

### 2.1. Study Design and Sampling

The research presented is based on the data obtained in a cross-sectional study performed among medical students of the Medical University of Silesia in Katowice between October 2019 and February 2022. A total of 1098 first-year medical students from three different academic years were recruited for this study. Therefore, it includes three groups of students tested in three consecutive research periods:Period before the COVID-19 pandemic—the study group includes 560 students from the academic year 2019/2020, where all classes were entirely conducted in a stationary form. All first-year medical students (*n* = 638) from the Faculty of Medical Sciences of the Medical University of Silesia in Katowice were invited to participate in the study, the response rate was 87.8%.The COVID-19 pandemic lockdown period—the study group includes 111 students from the academic year 2020/2021, where students attended all classes and exams in a remote form. The pandemic significantly hindered the research recruitment strategy of students for this academic year. Direct contact with students was impossible; hence, invitations sent via the Internet resulted in a significant decrease in the interest in study participation; the response rate was 22.6%.The COVID-19 pandemic period following lockdown—the study group includes 427 students of the academic year 2021/2022, where particular classes were conducted in a stationary form for practical training and a remote form for lectures. The response rate was high again and remains at the level of 83.7%.

All participants were requested to issue written consent to participate in the study. The project has the approval of the Bioethics Committee of the Medical University of Silesia in Katowice (approval number KNW/0022/KB/217/19; date: 8 November 2019).

### 2.2. Measurement Tools

The Quality of Life (QoL) of subjects was assessed with the Polish version of the WHOQOL-BREF questionnaire. The version of this questionnaire had been already validated in the Polish population [20] and we obtained approval from the World Health Organization (WHO) to use this instrument in our study. The WHOQOL-BREF was developed to assess the quality of life at the domain levels profile. It encompasses 26 items grouped in four QoL domains: somatic (physical health), psychological, social (social relationships) and environmental. Particular aspects of the quality of life were classified as satisfactory or problematic based on the respondents’ answers. The somatic domain includes energy and fatigue, mobility, pain and discomfort, the need for medical treatment, sleep and rest, as well as satisfaction with the capacity of work. The psychological domain includes satisfaction with the body, appearance and the frequency of positive and negative feelings in daily life. Questions on satisfaction with personal relationships, social support systems, and sexual activity are a part of the social relationships domain. The environmental domain includes questions on physical safety and security, availability of financial resources, accessibility and quality of social care and health services, home and physical environment satisfaction, opportunities for leisure time activities and also transport. According to the recommendation of WHO, the raw values were transformed into scores of from 0 to 100 points.

Furthermore, the AUDIT questionnaire was used to evaluate the prevalence of hazardous alcohol use among medical students. This tool was developed in 1989 by the WHO as a simple method of screening for excessive alcohol drinking. It consists of 10 questions on recent alcohol use, alcohol dependence symptoms and alcohol-related mental and physical problems [21]. Each response has a score ranging from 0 to 4; therefore, the range of possible scores is 0–40. Hazardous alcohol use was identified in the subjects with a score of ≥8 in AUDIT measurement, as mentioned in our previous paper [13].

In addition to the WHOQOL-BREF and AUDIT questionnaires, our final tool included questions on sociodemographic aspects (sex, age, current financial situation, current place of residence during medical studies), lifestyle (current traditional or electronic cigarette smoking, selected eating behaviors and frequency of physical activity), as well as health status indicators (chronic diseases, self-declared health status, BMI).

The research was based on a paper version of questionnaires used among students in the period before the COVID-19 pandemic and after the lockdown pandemic period, while the groups of students in the lockdown period received a digital version of the same tool. As we confirmed in the previous paper, both versions of questionnaires (paper and digital) represent excellent accordance and reproducibility [22].

### 2.3. Statistical Analysis

Initially, we presented the qualitative variables by the number of observations and percentages. Next, group differences for qualitative variables were tested by the Chi-square test. For quantitative variables, the data that identified particular QoL domains were originally analyzed using descriptive statistics such as mean values, standard deviation and ranges. Differences between groups were verified by the nonparametric test, the Kruskal–Wallis test with post-hoc comparisons as the distribution of variables does not meet the assumptions of normality. Finally, the statistically significant relationships between particular variables in bivariate analyses were tested using multivariable linear regression models. The dependent variables were the quality of life domains, while explaining variables were related to sociodemographic data, lifestyle, health status and particular study periods. The reference categories of all independent variables were coded as 1. The reliability of WHOQOL-BREF and AUDIT questionnaires were tested using Cronbach’s alfa coefficients. All domains had good internal consistency for *n* = 1098 (>0.7).

All analyses were performed in the Statistica 13.3 package (TIBCO Software Inc., Palo Alto, CA, USA), and *p* values below 0.05 were considered statistically significant.

## 3. Results

### 3.1. Personal Characteristics

The total number of participants was 1098. They were first-year medical students of the Medical University of Silesia in Katowice recruited in three different periods. The first study group included 560 students from the academic year 2019/2020, which was the time prior to the COVID-19 pandemic. In the second study group (the academic year 2020/2021), 111 students who practiced remote education due to the lockdown period were involved. The third study group of 427 students was involved in the research during the COVID-19 pandemic after the lockdown period (the academic year 2021/2022). Selected personal characteristics of the study groups including sociodemographic variables, data regarding lifestyle, health state and declared quality of life are shown in Table 1. Research participants more often reported themselves as female and single. Most of the respondents in all study groups declared a good financial situation and generally were living in a dormitory or rented flat. Furthermore, hazardous alcohol drinking was identified among approximately 30% of respondents and there were no statistically significant differences between particular groups (*p* = 0.08). On the other hand, statistically important relationships were indicated between the research period and the self-declared health status. We observed that 76.6% of students from the lockdown period (*n* = 111) reported their health as good, and this percentage was higher in comparison to the groups of students who were investigated before the pandemic and during the pandemic period after lockdown.

### 3.2. WHOQOL-BREF Scores

Table 2 presents obtained scoring of the QoL domains in particular groups of students recruited in different periods of the study. It was demonstrated that the overall QoL score was the highest among students from the lockdown period (M = 74.7; *p* = 0.002). This group of students also had the highest scores for the psychological and environmental domains of the QoL. It is worth mentioning that the observation among students who were examined during the pandemic period after lockdown indicated the highest scoring for the somatic domain (M = 62.8; *p* < 0.001). However, no statistically significant differences were determined for the social relationships domain (*p* = 0.3).

Additionally, Appendix A presents the results of post-hoc tests (Appendix A). It has been confirmed that the statistically significant differences between medical students from the period before the COVID-19 pandemic and those who were examined during the lockdown concerned somatic, psychological and environmental domains of quality of life.

### 3.3. Variables Related to the QoL Scores

Table 3 presents values of mean scores for the overall QoL of medical students in connection to a particular study period and selected independent variables. It was found that the variables significantly associated with lower scoring of the overall QoL were: poor financial situation, low frequency of physical activity and medically diagnosed chronic disease; this observation applies to all study periods. Furthermore, we observed that especially for students from the period before the COVID-19 pandemic, the overall QoL significantly depends on many factors such as marital status (*p* = 0.004), current place of residence (*p* < 0.001), hazardous alcohol drinking (*p* = 0.01), consumption of fruit and vegetables (*p* < 0.001) and BMI (*p* = 0.007).

In the final step, the total scoring for all QoL domains was verified using multivariable regression models (Table 4). Most importantly, we identified that the research period before the COVID-19 pandemic was the risk factor for the lower quality of life in terms of the somatic and environmental domains. It was also confirmed that determinants such as poor financial situation, low frequency of physical activity and bad self-declared health status harmed the QoL scores in all domains.

## 4. Discussion

Quality of life (QOL) is a broad multidimensional concept that usually includes subjective evaluations of both positive and negative aspects of life [23]. Health-related QoL (HRQoL) includes the physical, functional, social and emotional well-being of an individual [24]. Measurement of HRQOL can help determine the burden of preventable disease, injuries and disabilities, and can provide valuable new insights into the relationships between QoL and risk factors. It is especially important in the case of young people, medical students and future physicians who choose a very difficult job.

Current publication revealed that the COVID-19 pandemic has serious consequences on behavioral health and quality of life in the general population. The authors communicated that poorer mental health of subjects was related to the disruption of work, family and social life, as well as social isolation. Additionally, they explained that mental health problems were associated with a deterioration in quality of life [25]. Moreover, the findings of a study conducted in Italy confirmed the impact of the COVID-19 lockdown on the mental health deterioration of the total population. Specifically, it was shown that the rates of PTSD (post-traumatic stress disorder), symptoms of anxiety and depression, as well as insomnia were relatively high [26]. Some interesting observations were presented in a Polish and German population study, where it was indicated that quality of life, life satisfaction and well-being were affected by age, anxiety and risk of coronavirus infection. Older people declared better quality of life and experienced psychological functioning than younger respondents [27]. Quoted observations were contradictory with the results of a study conducted in Germany, in which most of the studied population declared no change in the quality of life compared to the period before the COVID-19 pandemic. Moreover, higher life satisfaction was associated with such factors as fewer mental health problems, higher income, no income loss during the pandemic, living with others and younger age [28].

The objective of the presented paper is to identify and assess determinants of the quality of life among medical students in one of the biggest medical universities in Poland with a simultaneous assessment of the COVID-19 pandemic’s impact on self-declared QoL.

Our findings revealed that better financial situation, higher physical activity and better self-declared health status were statistically significant factors having an impact on better quality of life of the first-year medical students at the MUoS in Katowice, Poland. This observation mostly corresponds with the results of the previous study among occupationally active inhabitants of the Silesian voivodeship aged 25–44 years old [29] in which it was confirmed that a higher score for the quality of life was associated with self-reported health status, levels of income, education, job satisfaction and frequency of physical activity. It is worth adding that socioeconomic factors were interlinked with inequalities of health among students in Germany [30]. For instance, a declared better social status was connected with better general, mental and physical health as well as higher physical activity. It cannot be debarred from the conclusions that this phenomenon might also involve Polish students; however, this should be verified in future analyses.

In addition, the level of QoL among medical students who had been examined before the COVID-19 pandemic period (*n* = 560) was lower in somatic, psychological and environmental domains than in the general population of young inhabitants of Silesian voivodeship [29]. Researchers from Norway and Canada have drawn similar conclusions, as their studies confirmed that medical students and physicians experienced a lower level of life satisfaction concerning population samples. It was found that concurrent situational factors, lack of social support and mental distress were of major importance [31,32]. What is more, a longitudinal study of medical students also showed that the level of life satisfaction decreased during the first year of studies and remained at this lower level until graduation [33]. The authors of the quoted study found that medical education and career have an unfavorable effect on the general life satisfaction of medical students and physicians and some results indicate that distress may correlate with impaired academic performance, cynicism, dishonesty, substance abuse and even suicide [33,34,35].

Nevertheless, the most important factor is to determine whether the COVID-19 pandemic affects the quality of life and health of medical students. Some authors documented that the prevalence of anxiety is similar to the period before the pandemic [36]. On the other hand, the results of meta-analyses suggest that the prevalence of depression and anxiety among dental students during the COVID-19 pandemic was high [37,38]. In another study conducted in Australia during the first period of the COVID-19 pandemic, most of the medical students participating in the research reported deterioration in mental well-being [5]. Furthermore, based on Malaysian observations, an impairment of the QoL level due to the COVID-19 pandemic was observed among medical university students. It concerned in particular the psychological and QoL domains, while the somatic and environmental scores were comparable to the pre-pandemic norms [39]. These observations are contradictory to the findings of our study, showing that the declared quality of life among medical students of the Medical University of Silesia in Katowice improved in the somatic and environmental domains during the COVID-19 pandemic period. It should be assumed that lockdown during the academic year was associated with a significant reduction in students’ cost of living (renting a flat, food costs, etc.), and studying at home (the form of online classes) significantly reduced the level of stress and ultimately resulted in better exam performance. However, the results of a study among first-semester medical students in the USA confirmed a 22.4% reduction in exam performance for the COVID cohort is reflective of significantly poorer performance across all five administered exams [40]. Results of another study among final-year medical students suggest a significant impact of COVID-19 on education; students felt less prepared for beginning work as a doctor [41]. It is worth noting that the results of our research are based on the comparison of pre-pandemic and pandemic data. Therefore, we believe that our findings should be correlated with the other research addressing the same parallels.

Some interesting observations were made in the cohort study of the first-year German medical students (*n* = 63), conducted in the period from entering medical university under ‘normal’ conditions to the COVID-19 situation and switching to online studies. Students were tested twice using a paper format questionnaire (in October and December 2019), and again twice during the virtual semester using a digital questionnaire (in June and December 2020). The results of this study showed a significant deterioration in mental health, including depression and burnout, mainly within the first two months of medical school. Then, the level of students’ well-being remained similar throughout the research period. It was generally concluded that the predominantly online learning during the COVID-19 pandemic did not contribute to any negative effects concerning the QoL and mental health of medical students. However, a greater psychological challenge for them was to adapt to the new situation of studying medicine at the university [42].

It seems that these observations are consistent with the results of the Croatian study on the impact of the first COVID-19 lockdown on the study satisfaction and burnout of medical students. This cross-sectional study includes information from 437 students collected before lockdown (in December 2019 and January 2020) and after (in June 2020). Additionally, there was a short six-month follow-up of 160 respondents conducted. Due to the COVID-19 pandemic, all universities in Croatia were closed, so the learning system was transformed into fully digital. An identical situation occurred in the German study and also in our country, Poland. Finally, the authors of this research did not observe any significant differences in burnout and study satisfaction among medical students before and after the full lockdown. Moreover, the levels of burnout were generally low during the whole follow-up period [43].

Another important issue corresponds to changes in lifestyle habits among medical students due to the COVID-19 pandemic. The findings of the study conducted in Croatia demonstrated longer sleep duration during the lockdown period, and fewer students reporting extreme tiredness and sleepiness when waking up in comparison to the pre-COVID-19 pandemic period [44]. It is worth adding that satisfaction with sleep and rest includes the somatic domain of the WHOQOL-BREF questionnaire that was used in our study. This may partially explain higher scores for Qol of medical students during the pandemic period.

In addition, there were no significant changes in dietary patterns between the periods that have been compared. The level of physical activity remained quite stable, and even a third of students from the lockdown period declared weight loss. Moreover, respondents declared a slightly lower average quality of life and a minor increase in stress perception during the lockdown in comparison to the pre-COVID period [44]. Our study also involved an assessment of selected aspects of lifestyle among medical students, so the results can be compared to the findings of the quoted research. We did not observe any prominent differences in the frequency of physical activity and eating behaviors between students examined before the COVID-19 pandemic and the pandemic period. Furthermore, the percentage of respondents with a BMI lower than 25.0 was the highest in the group of students from the COVID-19 pandemic lockdown period (*n* = 111). Nevertheless, these differences were not statistically significant. The last point concerns changes in the level of the quality of life among medical students during the COVID-19 pandemic period. We observed some methodological issues regarding the QoL measurement in the Croatian study. It should be noted that a very low number of questions were used to assess the QoL level among research participants instead of a validated tool that makes the assessment more reliable. In that case, the possibilities to compare the quality of life level among Croatian medical students to the results of our study are quite limited. In conclusion, we claim that the observations from German and Croatian studies may partially correspond with the finding of our research in the area of scoring in social relationships and the psychological domain of the QoL among medical students as it remained at a similar level during the COVID-19 lockdown period. Hence, there is still an outstanding matter—that is, clarifying why the QoL scoring in the somatic and environmental domain of students examined during the pandemic period was higher.

As with the last point, the influence of remote learning on the quality of life and health of medical students with the COVID-19 pandemic should be verified. The study conducted among dental students did not demonstrate a statistically significant negative effect with online learning due to the pandemic on their quality of life, anxiety and stress level [1]. On the other hand, an Italian study indicated a negative impact of remote learning on the mental health level of medical students, including symptoms of depression, during the COVID-19 outbreak [45]. Nevertheless, this research includes only one group of students (*n* = 233) who attended online classes during the first COVID-19 lockdown in Italy and there are no available data for the period before the pandemic.

Medical students’ perception of online studying during the COVID-19 pandemic seems worth considering in terms of their quality of life and health. A study conducted on a relatively large sample of Polish medical students (*n* = 804) proved that e-learning is a powerful tool for education. The advantages most frequently declared by students were the ability to stay at home, continuous access to materials, the opportunity to learn at their own pace and comfortable surroundings. They also reported some inconveniences, such as a lack of interactions with patients and technical problems with IT equipment, but there were no statistically significant differences observed between face-to-face and online learning in this study [46]. It was also found that online education can effectively complement the medical teaching process [47] and is useful in saving time and creating a flexible learning environment [48,49]. It was proved that even teaching surgical skills remotely is as effective and efficient as traditional medical education [50]. The results of these studies correspond with the data obtained from medical students examined in our study during the COVID-19 pandemic lockdown period (*n* = 111). This group of students most often declared living in a family home during studies at university (76%), which probably in combination with online education created a comfortable environment for learning during the pandemic. We believe that the above conclusions will guide us to understand why medical students who participated in our study declared a higher level of QoL during the COVID-19 pandemic in the somatic and environmental domains. The obtained results confirm that the immediate implementation of e-learning in the first year of medical studies, where theoretical subjects dominate, was a proper decision and had no impact on the deterioration of quality of life among students. In our opinion, this is an important observation that may be useful for university management teams who are responsible for teaching processes in potential future epidemic crises.

### Strengths and Limitations of This Study

The main limitation of our study was the low response rate (22.6%) in the group of medical students recruited during the lockdown period of the COVID-19 pandemic. Nevertheless, statistical modeling was used in the regression analyses by combining this study group with a group of students from the COVID-19 pandemic period after lockdown (*n* = 427), where the participation rate was significantly higher (83.7%). Finally, the total number of participants included in the research period during the pandemic was 538. It is worth noting that the group size of medical students from the period before the COVID-19 pandemic was similar (*n* = 560), which is a strong base of the research from a statistical point of view. However, despite efforts to eliminate selection bias, we are aware that the results obtained in a cross-sectional study are not sufficient for unambiguous conclusions. For this reason, we will continue our project in the next years as a cohort study. We will test students from each group in question twice more, during the second and fifth years of studies. Such a scenario will allow us to assess how QoL will change in these three independent groups over time.

## 5. Conclusions

In conclusion, the obtained results confirmed that better financial situation, higher physical activity and better self-declared health status were statistically significant factors improving the quality of life of first-year medical students in Poland. The findings of our study also showed that the declared somatic and environmental domains of QoL among medical students were better during the COVID-19 pandemic period. Our observations confirmed that the immediate implementation of e-learning could protect against the deterioration of mental health and quality of life in first-year medical students during possible future epidemic crises.

## Figures and Tables

**Table 1 ijerph-19-11888-t001:** Selected personal characteristics of the study groups and *p*-values of the Chi-square test.

Variable		Period before the COVID-19 Pandemic (2019/2020) *n* = 560		The COVID-19 Pandemic Lockdown Period (2020/2021) *n* = 111		The COVID-19 Pandemic Period after Lockdown (2021/2022) *n* = 427		Results of Chi^2^ Test
		N	%	N	%	N	%	
Sex	Women	342	61.1	70	63.1	289	67.7	chi^2^ = 4.61 *p* = 0.09
	Men	218	38.9	41	36.9	138	32.3	
Marital status	In relationship	157	28.0	28	25.2	99	23.2	chi^2^ = 2.61 *p* = 0.2
	Single	399	71.3	83	74.8	319	74.7	
	Missing data	4	0.7	0	0.0	9	2.1	
Current financial situation	Poor	138	24.6	20	18.0	115	26.9	chi^2^ = 3.77 *p* = 0.1
	Good	422	75.4	91	82.0	312	71.3	
Current place of residence during studies at university	Family home	142	25.4	37	33.3	124	29.0	chi^2^ = 3.67 *p* = 0.1
	Dormitory/rented flat or room	418	74.6	74	66.7	303	71.0	
Current cigarette smoking (traditional or electronic)	Yes	106	18.9	26	23.4	100	23.4	chi^2^ = 3.38 *p* = 0.1
	No	454	81.1	85	76.6	326	76.3	
	Missing data	0	0.0	0	0.0	1	0.2	
Hazardous alcohol use	Hazard use	167	29.8	32	28.8	128	30.0	chi2 = 0.37 *p* = 0.8
	Low risk	373	66.6	79	71.2	274	64.2	
	Missing data	20	3.6	0	0.0	0	0.0	
Number of meals containing animal protein	100% of meals	72	12.9	11	9.9	53	12.4	chi^2^ = 10.41 *p* = 0.03
	75% of meals	298	53.2	62	55.9	191	44.7	
	Less often	190	33.9	38	34.2	183	42.9	
Consumption of fruit and vegetables	Daily (≥3 meals)	95	17.0	20	18.0	85	19.9	chi^2^ = 2.74 *p* = 0.6
	Daily (≥2 meals)	274	48.9	59	53.2	198	46.4	
	Less often	189	33.8	32	28.8	142	33.3	
	Missing data	2	0.4	0	0.0	2	0.5	
Frequency of physical activity	High	459	82.0	88	79.3	347	81.3	chi^2^ = 0.44 *p* = 0.7
	Low	101	18.0	23	20.7	79	18.5	
	missing data	0	0.0	0	0.0	1	0.2	
Self-declared health status	Bad	228	40.7	26	23.4	175	41.0	chi^2^ = 12.76 *p* = 0.001
	Good	332	59.3	85	76.6	251	58.8	
	Missing data	0	0.0	0	0.0	1	0.2	
Ever diagnosed with chronic disease	Yes	123	22.0	26	23.4	101	23.7	chi^2^ = 0.36 *p* = 0.8
	No	432	77.1	85	76.6	324	75.9	
	Missing data	5	0.9	0	0.0	2	0.5	
Declared quality of life	High	402	71.8	91	82.0	310	72.6	chi^2^ = 5.00 *p* = 0.08
	Low	158	28.2	20	18.0	117	27.4	
	Missing data	0	0.0	0	0.0	0	0.0	
Body mass index (BMI)	≤24.9	468	83.6	99	89.2	371	86.9	chi^2^ = 12.91 *p* = 0.2
	≥25.0	89	15.9	12	10.8	56	13.1	
	Missing data	3	0.5	0	0.0	0	0.0	

**Table 2 ijerph-19-11888-t002:** Summary of the WHQOL-BREF domains (scoring after transformation).

Quality of Life Domain	Period before the COVID-19 Pandemic (2019/2020) *n* = 560		The COVID-19 Pandemic Period during Lockdown (2020/2021) *n* = 111		The COVID-19 Pandemic Period after Lockdown (2021/2022) *n* = 427		Results of K–W Test
	M (SD)	Range (Min–Max)	M (SD)	Range (Min–Max)	M (SD)	Range (Min–Max)	
Overall QoL	68.9 (18.1)	0.00–100.0	74.7 (15.9)	25.0–100.0	68.6 (17.9)	12.5–100.0	H = 12.45 *p* = 0.002
Somatic	43.2 (12.6)	7.1–75.0	59.5 (8.3)	39.3–71.4	62.8 (15.2)	21.4–96.4	H = 378.31 *p* < 0.001
Psychological	60.8 (13.2)	16.7–95.8	64.7 (12.9)	33.3–83.3	62.2 (16.8)	12.5–100.0	H = 53.15 *p* = 0.001
Social relationships	69.9 (20.4)	8.3–100.0	72.1 (19.3)	25.0–100.0	69.5 (18.6)	0.0–100.0	H = 16.50 *p* = 0.3
Environmental	64.0 (13.7)	25.0–96.9	72.1 (19.3)	25.0–100.0	64.4 (12.6)	21.9–93.8	H = 54.51 *p* < 0.001

M, mean; SD, standard deviation; Min, Minimum; Max, Maximum; K–W, Kruskal–Wallis test; *p*, statistical significance.

**Table 3 ijerph-19-11888-t003:** Scoring for the Overall Quality of Life in medical students according to the study period and particular independent variables.

Independent Variable		Overall QoL					
		Period before the COVID-19 Pandemic (2019/2020) *n* = 560		The COVID-19 Pandemic Period during Lockdown (2020/2021) *n* = 111		The COVID-19 Pandemic Period after Lockdown (2021/2022) *n* = 427	
		M (SD)	Statistical Significane	M (SD)	Statistical Significane	M (SD)	Statistical Significane
Sex	Women	69.3 (17.7)	Z = −0.16 *p* = 0.8	72.5 (15.6)	Z = −2.07 *p* = 0.03	67.3 (18.2)	Z = −1.82 *p* = 0.06
	Men	68.2 (18.8)		78.4 (16.1)		71.3 (17.0)	
Marital status	In relationship	65.7 (17.2)	Z = −2.85 *p* = 0.004	72.8 (17.9)	Z = 0.17 *p* = 0.8	67.8 (21.5)	Z = 0.07 *p* = 0.9
	Single	70.0 (18.3)		75.3 (14.9)		68.8 (16.8)	
Current financial situation	Poor	61.1 (18.5)	Z = 5.28 *p* < 0.001	64.4 (16.9)	Z = 2.81 *p* = 0.004	64.2 (18.0)	Z = 2.72 *p* = 0.006
	Good	71.4 (17.3)		76.9 (14.9)		70.2 (17.7)	
Current place of residence during studies at university	Family home	73.8 (18.6)	Z = −3.77 *p* < 0.001	76.0 (13.0)	Z = −0.28 *p* = 0.7	69.3 (18.3)	Z = 0.60 *p* = 0.6
	Dormitory/rented flat or room	67.2 (18.3)		74.0 (17.3)		68.3 (17.8)	
Current cigarette smoking (traditional or electronic)	Yes	67.0 (18.5)	Z = −1.16 *p* = 0.2	72.6 (17.0)	Z = 0.71 *p* = 0.4	59.7 (17.1)	Z = 5.76 *p* < 0.001
	No	69.3 (18.0)		75.3 (15.7)		71.2 (17.3)	
Hazardous alcohol use	Low risk	69.9 (18.5)	Z = −2.44 *p* = 0.01	75.3 (16.1)	Z = 0.52 *p* = 0.6	68.3 (19.0)	Z = 0.75 *p* = 0.4
	Hazard use	65.9 (17.0)		73.0 (15.6)		67.8 (16.0)	
Number of meals containing animal protein	100% of meals	70.2 (18.9)	H = 1.99 *p* = 0.3	84.1 (13.8)	H = 4.68 *p* = 0.9	75.7 (15.6)	H = 9.45 *p* = 0.008
	75% of meals	69.4 (17.7)		73.2 (15.2)		68.3 (17.4)	
	Less often	67.5 (18.5)		74.3 (17.2)		66.7 (18.6)	
Consumption of fruit and vegetables	Daily (≥3 meals)	74.9 (18.1)	H = 22.22 *p* < 0.001	73.8 (14.0)	H = 3.22 *p* = 0.1	71.2 (15.8)	H = 2.60 *p* = 0.2
	Daily (≥2 meals)	69.9 (16.9)		77.1 (15.8)		68.6 (17.9)	
	Less often	64.6 (18.7)		70.7 (17.0)		67.4 (18.7)	
Frequency of physical activity	High	70.1 (17.5)	Z = 3.35 *p* < 0.001	76.8 (14.5)	Z = 2.64 *p* = 0.008	69.7 (17.4)	Z = −2.60 *p* = 0.009
	Low	63.4 (19.9)		66.3 (18.6)		63.6 (19.3)	
Ever diagnosed with chronic disease	Yes	68.3 (18.0)	Z = 3.31 *p* < 0.001	65.4 (15.1)	Z = 3.35 *p* < 0.001	62.1 (19.4)	Z = 3.98 *p* < 0.001
	No	70.3 (17.8)		77.5 (15.2)		70.5 (17.0)	
Body mass index (BMI)	≤24.9	69.8 (17.9)	Z = 2.66 *p* = 0.007	75.1 (15.8)	Z = 0.94 *p* = 0.3	69.2 (17.6)	Z = 1.82 *p* = 0.06
	≥25.0	64.5 (18.5)		70.8 (17.1)		64.3 (19.6)	

*n*, number of participants; M, mean; SD, standard deviation; Z, results of the U Mann–Whitney test; H, results of Kruskal–Wallis test; *p*, statistical significance.

**Table 4 ijerph-19-11888-t004:** Results of multivariable linear regression models for the relationship between QoL domains and particular independent variables (*n* = 1098).

Independent Variable	Regression Coefficient (95% CI ^a^)	*p* ^b^
Overall QoL (*n* = 1098, R^2 c^ = 0.13, *p* < 0.001 ^d^)		
Marital status (1 = in relationship, 2 = single)	0.05 (0.00, 0.11)	0.04
Current financial situation (1 = poor, 2 = good)	0.20 (0.14, 0.25)	<0.001
Current place of residence during studies at university (1 = family home, 2 = dormitory/rented flat or room	−0.08 (−0.14, 0.02)	0.004
Current traditional or electronic cigarettes smoking (1 = yes, 2 = no)	0.10 (0.04, 0.15)	<0.001
Frequency of physical activity (1 = high, 2 = low)	−0.13 (−0.16, −0.04)	<0.001
Ever diagnosed chronic disease (1 = yes, 2 = no)	0.16 (0.11, 0.22)	<0.001
Somatic (*n* = 1098, R^2 c^ = 0.46, *p* < 0.001 ^d^)		
Sex (1 = women, 2 = men)	0.06 (0.01, 0.10)	0.01
Current financial situation (1 = poor, 2 = good)	0.05 (0.01, 0.10)	0.01
Hazardous alcohol use (1 = low risk, 2 = hazard use)	−0.05 (−0.09, −0.005)	0.02
Frequency of physical activity (1 = high, 2 = low)	−0.11 (−0.15, −0.06)	<0.0001
Ever diagnosed chronic disease (1 = yes, 2 = no)	0.05 (0.007, 0.10)	0.02
Self-declared health status (1 = bad, 2 = good)	0.26 (0.21, 0.31)	<0.001
Research period (1 = before pandemic, 2 = pandemic period)	0.57 (0.53, 0.62)	<0.001
Psychological (*n* = 1098, R^2 c^ = 0.19, *p* < 0.001 ^d^)		
Sex (1 = women, 2 = men)	0.06 (0.002, 0.11)	0.04
Current financial situation (1 = poor, 2 = good)	0.09 (0.04, 0.15)	<0.001
Hazardous alcohol use (1 = low risk, 2 = hazard use)	−0.08 (−0.13, −0.02)	0.005
Frequency of physical activity (1 = high, 2 = low)	−0.09 (−0.15, −0.03)	0.001
Ever diagnosed chronic disease (1 = yes, 2 = no)	0.06 (0.004, 0.11)	0.03
Self-declared health status (1 = bad, 2 = good)	0.35 (0.29, 0.41)	<0.001
Social relationships (*n* = 1098, R^2 c^ = 0.12, *p* < 0.001 ^d^)		
Marital status (1 = in relationship, 2 = single)	−0.20 (−0.25, −0.14)	<0.001
Current financial situation (1 = poor, 2 = good)	0.10 (0.04, 0.16)	<0.001
Frequency of physical activity (1 = high, 2 = low)	−0.07 (−0.12, 0.01)	0.01
Self-declared health status (1 = bad, 2 = good)	0.22 (0.16, 0.28)	<0.001
Environmental (*n* = 1098, R^2 c^ = 0.26, *p* < 0.001 ^d^)		
Current financial situation (1 = poor, 2 = good)	0.22 (0.16, 0.27)	<0.001
Current place of residence during studies at university (1 = family home, 2 = dormitory/rented flat or room	−0.09 (−0.14, −0.04)	<0.001
Frequency of physical activity (1 = high, 2 = low)	−0.08 (−0.14, −0.03)	0.001
Ever diagnosed chronic disease (1 = yes, 2 = no)	0.07 (0.01, 0.12)	0.009
Self-declared health status (1 = bad, 2 = good)	0.35 (0.29, 0.41)	<0.001
Research period (1 = before pandemic, 2 = pandemic period)	0.05 (0.002, 0.10)	0.03

^a^ CI, Confidence Interval. ^b^
*p*, significance to the reference group. ^c^ R^2^, determination of the model. ^d^
*p*, the significance of the multivariable regression model. The reference group was coded as 1.

## Data Availability

The data presented in this study are available on reasonable request from the corresponding author. The data are not publicly available due to data sensitivity and to protect the interests and privacy of the respondents.

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
