# Peer review of "The Association between COVID-19 Pandemic and the Quality of Life of Medical Students in Silesian Voivodeship, Poland"

_ijerph, 2022, doi:10.3390/ijerph191911888_

Round 1

Reviewer 1 Report

This work will bring about awareness of the impact of Covid-19 on the quality of life of students in Poland and identify further support. Some areas for improvement of the manuscript:

Title: Please avoid the abbreviation QoL

This is a cross-sectional study, "impact" should be replaced by "the association"

Abstract: please modify "the COVID-19" not "The COVID-19"

Introduction:

Please clarify the impact of the covid-19 on the first-year medical student due to the characteristics of selected participants.

Methods

1. Participants:

- Please mention the reason for choosing the Medical University of Silesia in Katowice and first-year students

- background of selected medical students (medical doctor, nursing or other major etc)

- eligible and exclusion criteria for selecting participants

did you do sample size estimation?

- Which digital platform was used to collect data and how to control the quality of answers.

2. Data analysis

- which multivariable linear regression model was applied, have you tested the normal distribution of the dependent variable

- please describe how to select the independent variables in the multivariate regression model for overall Qol and each factor.

Result

- Table 4, please mention reference group

Discussion

- Paragraph 2, you mentioned "QoL prior to the COVID-19 pandemic period was lower in somatic, psychological, and environmental domains" and "What is more, a six-year longitudinal study of medical students showed that the level of life satisfaction decreased during the first year of studies and remained at this lower level until graduation". 

The reference study suggested changes in life satisfaction of medical students during six-year. However, your study was cross-sectional, before pandemic and pandemic period were examined in different students. Thus, the explanation is not appropriate.

- Please explain more on the different of QoL in different period as impact of Covid-19, what are advantages and disadvantages.

- Please provide the implication of study. What should recommended for medical students if another pandemic occur like Covid-19

Overall comment:

- The author needs to have the paper checked by a native speaker.

Author Response

Dear Reviewer,

We are very grateful for your time that you spent for the review of our manuscript and your valuable comments and suggestions. Please see our explanations in the attached file.

Regards,

Szymon Szemik

Reviewer 2 Report

Dear Editor,

Thank you for this opportunity to reivew this manuscript entitled "The Impact of COVID-19 Pandemic on the QoL of Medical Students in Silesian Voivodeship, Poland". I have just finished my review and I must declare that I enjoyed while reading this manuscript. The problems of this manuscript are mainly about methodology.

COVID-19 researches are still trendy and the topic of this manuscript is interesting. Examining the determinants of the  quality of life among medical students in Poland before and during the COVID-19 pandemic can present a new perspective for the new COVID-19 researches.

The strong side of the mansucript is that the researchers collected data from three different groups in different time intervals. That means the findings can be comperable. However, the number of participants in the groups is uneven. I am happy to see that this drawback was expressed in the limitations section. The authors wrote on page 2 and 3 that " First  study group includes 560 students from  the academic year 2019/2020, that means the response rate was 87.8%. Second group includes 111 students from  the academic year 2020/2021, that means the response rate was  22.6%. And third group includes 427 students of the academic year 2021/2022,  that means the response rate was  83.7%.  I just recommend the authors to exclude the second group and repeat the anlaysis again. Because I think the data (as the authors stated) did not have normal distribution because of the second group.  

In the manuscript I could not see the Cronbach’s alpha results (reliability). Please add the reliability analysis results. 

The authors stated on page 8 that "In the final step, the total scoring for all QoL domains was verified using multivariable regression models". Before conducting the analysis for multivariable regression model, the assumptions of multivariable linear regression should be checked (For example linear relationship, no multicollinearity etc.). Please make you sure that these assumptions are met and then include the finding into your manuscript before the regression analysis.

The discussion part should be developed and strenghtened. There is a huge literature about COVID-19 and Quality of Life from very different contexts. Please discuss your findings in detail. 

My concluding remarks are about the implications of this study. We have completed this research and reported. According to this specific research results, what should university management teams do, what should the policy makers do? Please include concrete recommendations and implications into your paper based on your research findings.

Author Response

(The authors gave the same response as above.)

Round 2

Reviewer 2 Report

In my first review I strongly recommended the  authors to strenghten and extend the discussion, conclusion and implications sections. Unfortunately the authors did almost nothing with these parts. In revised paper the authhors only added two new references but it is not enough for this manuscript. We have a huge COVID-19 literature and the authors should have benefited from this.

The paper seems very insufficient to be published in IJERPH in its current version.

Regards.

Author Response

Dear Reviewer,

Thank you again for your time and effort you put into reviewing our manuscript. We believe that changes we made in the Discussion section will meet your positive feedback. 

Regards,

Szymon Szemik
